# Genes Located on 18q23 Are Epigenetic Markers and Have Prognostic Significance for Patients with Head and Neck Cancer

**DOI:** 10.3390/cancers11030401

**Published:** 2019-03-21

**Authors:** Kiyoshi Misawa, Takeharu Kanazawa, Daiki Mochizuki, Atsushi Imai, Masato Mima, Satoshi Yamada, Kotaro Morita, Yuki Misawa, Kazuya Shinmura, Hiroyuki Mineta

**Affiliations:** 1Department of Otolaryngology/Head and Neck Surgery, Hamamatsu University School of Medicine, Hamamatsu, Shizuoka 431-3192, Japan; daiki_m525@yahoo.co.jp (D.M.); imaimimi@yahoo.co.jp (A.I.); tendoon@gmail.com (M.M.); veridique.star@gmail.com (S.Y.); drb94amm1307@yahoo.co.jp (K.M.); mswyuki@abox3.so-net.ne.jp (Y.M.); mineta@hama-med.ac.jp (H.M.); 2Department of Otolaryngology, Tokyo Voice Center, International University of Health and Welfare, Tokyo 107-0052, Japan; mayuinannarbor@yahoo.co.jp; 3Department of Tumor Pathology, Hamamatsu University School of Medicine, Hamamatsu, Shizuoka 431-3192, Japan; kzshinmu@hama-med.ac.jp

**Keywords:** head and neck cancer, LOH, 18q23, CpG island, promoter methylation, quantitative methylation-specific PCR, disease-free survival

## Abstract

Loss of heterozygosity (LOH) on chromosome 18q23 is associated with significantly decreased survival in head and neck cancer. In agreement with such tumor suppressive roles, the loss of function of genes located in this region can be achieved through LOH and promotor hypermethylation. In this study, the methylation status of promoters of 18q23 genes in 243 head and neck cancer patients was assessed by quantitative methylation-specific PCR. Promoter methylation was then compared to various clinical characteristics and patient survival. *GALR1* and *SALL3* promoter methylation correlated with reduced disease-free survival (log-rank test, *p* = 0.018 and *p* = 0.013, respectively). Furthermore, based on multivariate Cox proportional hazards analysis, these methylation events were associated with poor disease-free survival, with hazard ratios of 1.600 (95% confidence interval: CI, 1.027–2.493; *p* = 0.038) and 1.911 (95% CI, 1.155–3.162; *p* = 0.012), respectively. By comparison, *GALR1* and *SALL3* methylation were not prognostic for overall survival in The Cancer Genome Atlas (TCGA) cohort. Our findings suggest that the methylation status of 18q23 genes could serve as important biomarkers for the prediction of clinical outcomes in well-annotated head and neck squamous cell carcinoma cohorts. *GALR1* and *SALL3* methylation could thus help to facilitate risk stratification for individualized treatment.

## 1. Introduction

In head and neck squamous cell carcinoma (HNSCC), epigenomic inactivation linked to tumor suppressor genes (TSGs) is more frequent compared to somatic mutations in cancer and might drive tumorigenic initiation and progression [1]. Several studies have used gene-specific and genome-wide approaches to examine epigenetic changes and methylation in HNSCC [2,3]. It is widely accepted that understanding such epigenetic events is critical to elucidate the mechanisms through which environmental factors contribute to the development and progression of human cancer [4]. Increasing evidence supports the existence of specific epigenetic changes related to associated risk-factor exposure, which might be exploited to discover new biomarkers that could ultimately improve the prevention, diagnosis, and treatment of cancer [5].

Clinical staging and histological grading systems are convenient but imperfect predictors of recurrence. Risk factors for HNSCC development include tobacco and alcohol abuse, sexual promiscuity, and human papilloma virus (HPV) infection [6]. Patients with HPV-associated oropharyngeal SCC (OPSCC) have a better prognosis than patients with HPV-negative tumors when treated with multimodal therapies [7,8]. Thus, there is a need to further refine and apply concerted approaches to molecular biomarkers for HNSCC, which might help to facilitate early detection, improved monitoring, and the delivery of individualized cancer therapy.

Loss of heterozygosity (LOH) on chromosome 18q is associated with significantly decreased survival in HNSCC [9]. The commonly lost region of 18q is 18q23, which is lost in 53% (D18S461) to 75% (D18S70) of cases [10]. Our preliminary analyses indicated that methylation-induced galanin receptor type 1 (*GALR1*) gene silencing is a critical event in HNSCC progression and that restoring expression inhibits tumor cell growth [11]. Furthermore, transcriptional inactivation of Sal-like protein 3 (*SALL3*) was found to be associated with aberrant methylation of other tumor-related genes and ten-eleven translocation (*TET*) family genes, as well as DNA methyltransferase 3 alpha (*DNMT3A*) levels, in HNSCC [12]. These findings are consistent with the notion that the methylation status and inactivation of other 18q23 genes might contribute to aggressive tumor behavior in HNSCC.

To test this hypothesis, we investigated the methylation status of 18q23 genes based on two prospective cohort studies (original cohort of 243 patients and The Cancer Genome Atlas (TCGA) comprising 516 cases). We assessed such epigenetic events at diagnosis and during follow-up to assess their clinical significance and potential as prognostic biomarkers for tumor recurrence and patient survival. Furthermore, we carefully reviewed the literature with respect to the association between the methylation of 18q23 genes and survival in patients with human cancers. To our knowledge, this study is the first to implicate 18q23 gene methylation in the prognosis of HNSCC.

## 2. Results

### 2.1. Initial Screening: Promoter Methylation Status of 12 Genes Located on 18q23 in Matched Pairs of Head and Neck Tumors and Adjacent Normal Mucosal Tissues

Initially, the promoter methylation status of 12 genes was analyzed in 36 cancerous and paired noncancerous mucosae by quantitative methylation-specific PCR (Q-MSP). Promoter methylation levels were represented by normalized methylation values (NMVs), which are the ratios of methylated DNA at the target sequence in each specimen to fully methylated control DNA. We noted distinct methylation patterns as follows: (a) a higher frequency and quantity of methylated genes was noted in HNSCCs compared to that in controls with the absence of methylation in a subset of normal samples (*p* < 0.05), including ATPase phospholipid transporting 9B (*ATP9B*), *GALR1*, potassium voltage-gated channel modifier subfamily G member 2 (*KCNG2*), par-6 family cell polarity regulator gamma (*PARD6G*), and *SALL3*; (b) a higher frequency of methylated genes was noted in HNSCC compared to that in controls but with similar quantities in both groups (*p* < 0.10), including heat shock factor binding protein 1 like 1 (*HSBP1L1*) and nuclear factor of activated T cells 1 (*NFATC1*); (c) methylation of genes was noted in both HNSCC and controls at a similar frequency with no difference in methylation levels (*p* ≥ 0.10), including CTD phosphatase subunit 1 (*CTDP1*), PQ loop repeat containing 1 (*PQLC1*), ribosome binding factor A (*RBFA*), thioredoxin like 4A (*TXNL4A*), and zinc finger protein 516 (*ZNF516*) (Figure 1a and Appendix A). Based on these results, seven genes (*ATP9B*, *GALR1*, *HSBP1L1*, *KCNG2*, *NFATC1*, *PARD6G* and *SALL3*) could distinguish HNSCC samples from normal samples, and these were selected for further testing using the expanded cohort.

### 2.2. Promoter Methylation Status of Seven Genes in Original and TCGA Cohorts

Q-MSP was next used to assess the promoter methylation status of seven genes located on 18q23 in 243 primary HNSCC samples. The methylation rates for these genes were as follows: *ATP9B*, 37.0%; *GALR1*, 49.4%; *HSBP1L1*, 30.5%; *KCNG2*, 59.3%; *NFATC1*, 9.1%; *PARD6G*, 40.3% and *SALL3*, 66.3% in original cohort. In the TCGA cohort, we also detected frequent methylation of *ATP9B* (33.7%), *GALR1* (95.9%), *HSBP1L1* (39.0%), *KCNG2* (68.6%), *NFATC1* (70.0%), *PARD6G* (28.9%) and *SALL3* (85.9%) in HNSCC (Figure 1b). The mean number of methylated genes per sample (original cohort and TCGA cohort) was 2.72 (range, 0–6) and 4.22 (range, 1–7), respectively. In the original cohort, at least one of these genes was methylated in most samples (231 of 243 samples, 95.1%). In the TCGA cohort, aberrant methylation of at least one of the seven genes was detected in all cases (Figure 1c). Regarding the relationships between mRNA expression and DNA methylation, Pearson correlation analysis revealed significant inverse correlations for *GALR1* and *SALL3*, positive correlations for *HSBP1L1*, and no correlations of *ATP9B*, *KCNG2*, *NFATC1* and *PARD6G* in the TCGA cohort (Appendix A). From the TCGA database, the average β values for *GALR1*, *NFATC1* and *SALL3* methylation were significantly higher in the HNSCC samples than in the normal samples (*p* < 0.001). Furthermore, the methylation of *ATP9B*, *HSBP1L1*, *KCNG2* and *PARD6G* promoters was not associated with HNSCC and normal control groups (Appendix A).

### 2.3. Correlation between the Methylation Status of Seven Gene Promoters Located on 18q23 and Clinicopathological Parameters

The associations between the methylation status of target genes and clinicopathological features in the original cohort are summarized in Table 1. A higher recurrence rate was observed for patients with methylated *GALR1* (*p* = 0.007). Methylation of *SALL3* was also associated with recurrence events (*p* = 0.033) (Table 1). In the TCGA cohort, methylation of the *ATP9B* promoter was significantly correlated with tumor size (*p* = 0.023) and clinical stage (*p* = 0.006), methylation of the *GALR1* promoter was significantly correlated with lymph node status (*p* = 0.024), and methylation of the *HSBP1L1* promoter was significantly correlated with sex (*p* < 0.001) and smoking status (*p* = 0.002) (Table 2). Continuous marker methylation analyses showed no association between the methylation index (MI) of any of the seven target genes and age at disease onset, sex, alcohol consumption, smoking status, tumor size, lymph node status, clinical stage, or recurrence (Appendix A).

### 2.4. Kaplan–Meier Analysis

Based on the original cohort, Kaplan–Meier survival curves for each of the seven genes are shown in Figure 2. Disease-free survival (DFS) time did not significantly differ between patients with methylated genes and those with unmethylated genes, with two notable exceptions; this was significantly shorter when *GALR1* was methylated (*p* = 0.018; Figure 2b) and when *SALL3* was methylated (*p* = 0.013; Figure 2g). For *GALR1* and *SALL3*, the DFS rates for both unmethylated genes, either methylated genes, and both methylated genes, were 66.9%, 59.5% and 0%, respectively (*p* = 0.010; Figure 2h). Based on log-rank tests, a trend in poorer DFS for patients with the methylation phenotype, defined as ≥5 methylated genes, was observed (*p* = 0.056; Appendix A). To validate the prognostic implications of the methylation of the seven target genes, we examined the data for the 507 HNSCC patients in TCGA database. Methylation of any gene was not associated with an altered overall survival rate when compared to that for samples harboring low levels of methylation (Appendix A).

### 2.5. Prognostic Value of the Methylation Status of Seven Genes Located on 18q23

The association between methylation and risk of recurrence was estimated via multivariate analysis using a Cox proportional hazards model adjusted for age, HPV status, smoking status, alcohol consumption, and clinical stage. In patients exhibiting *GLAR1* promoter methylation (120/243, 49.4%), the adjusted odds ratio (OR) for recurrence was 1.600 (95% confidence interval [CI]: 1.027–2.493, *p* = 0.038). *SALL3* methylation (161/243, 66.3%) also showed a significant association with recurrence (OR = 1.911, 95% CI: 1.155–3.162, *p* = 0.012; Table 3).

## 3. Discussion

The identification of epigenetic modifications of 18q23 genes is important to elucidate the mechanisms underlying tumorigenesis and to assess recurrence risk; we studied this in 243 HNSCC patients. We found that aberrant methylation of *GALR1* and *SALL3* promoters was positively correlated with recurrence in patients with HNSCCs. To our knowledge, this study is the first to show that the epigenetic regulation of 18q23 genes can provide insights into the aggressive tumor behavior and the risk of disease recurrence in HNSCC.

It was reported that the loss of chromosome 18q develops with tumor progression and is associated with significantly decreased survival in HNSCC patients [10,13]. The missing portion of 18q23 can vary from 53% (D18S461) to 75% (D18S70) and encompasses the *GALR1* and *SALL3* locus [10]. Somatic gene mutations are not the only mechanism of biallelic inactivation. Loss of 18q occurs frequently in HNSCC, with the loss of 18q23 reported in 55% of HNSCCs in the TCGA database [14]. In laryngeal carcinoma, loss of heterozygosity at 18q23 was found to be associated with lymph node involvement and worse prognosis [15].

Interestingly, we found that aberrant methylation of *GALR1* and *SALL3* is associated with worse DFS and that this might be a critical event in HNSCC. Based on the TCGA cohort of HNSCC, *GALR1* and *SALL3* mutations occurred at low frequencies, in three (0.58%) and nine (1.75%) of 515 patients, respectively [16]. Furthermore, for *GALR1* and *SALL3*, significant inverse correlations were found between mRNA expression and DNA methylation in this cohort. Furthermore, for the TCGA cohort, we found that *GALR1* and *SALL3* were the best prognostic markers for patients with HNSCC. Our findings provide evidence that *GALR1* and *SALL3* methylation might represent a good biomarker to predict HNSCC recurrence. This could facilitate HNSCC screening and the development of surveillance programs.

*GALR1* is one of three G-protein coupled receptors (GPCRs) for galanin, a neuropeptide encoded by the *GALR1* gene that is widely expressed in several peripheral tissues including the gastrointestinal tract, skeletal muscle, heart, kidney, uterus, ovary, and testis, in addition to the central nervous system [17]. GALR1-transfected HNSCC cells demonstrated decreased cell proliferation and colony formation after galanin stimulation [18]. Thus, *GALR1* might represent a tumor suppressor in HNSCC. Recently, it was reported that aberrant *GALR1* promoter methylation is significantly associated with shortened survival in salivary duct carcinoma patients [19]. The presence of *GALR1* methylation in vaginal swabs also indicates the presence of endometrial malignancy with a sensitivity of 92.7% and a specificity of 78.9% [20]. Therefore, *GALR1* DNA methylation is one common molecular alteration in human cancers (Table 4).

The product of *NFATC1* is a component of the nuclear factor of activated T cells DNA-binding transcription complex. Hypomethylation of the promoter region of *NFATC1* was identified in chronic lymphocytic leukemia patients and found to correlate with disease stage [21]. Strong *NFATC1* expression was also found to be significantly associated with worse patient outcomes in urothelial carcinoma [22]. Furthermore, *NFATC1* overexpression in high-grade serous ovarian carcinoma is an independent prognostic factor of poor overall survival and early relapse [23]. *NFATC1* might thus play a causative role in oncogenesis (Table 4). In HNSCC, *NFATC1* comprises a region that still needs to be explored. Our results indicated that *NFATC1* hypermethylation occurs with extremely low frequency and is not associated with outcome.

Vertebrate spalt proteins are classified into four groups, encoded by four genes in humans, namely *SALL1–4* [24]. Four genes have also been identified in mice, namely Sall1–4, whereas three have been found in chickens, three in zebrafish, and five in xenopus [24]. *SALL3* is an epigenetic hotspot for aberrant DNA methylation and is associated with abnormal placental development in mice [25]. Several clinical studies have been carried out to investigate the association between *SALL3* and carcinogenesis. Shikauchi et al. reported that *SALL3* can be silenced by DNA methylation and that the encoded protein interacts with DNA methyltransferases 3 alpha (DNMT3A) in hepatocellular carcinoma [26]. Aberrant hypermethylation of *SALL3* is also positively associated with HPV infection in cervical cancer [27]. Furthermore, recurrent bladder cancer patients were found to display higher proportions of *SALL3* DNA methylation compared to that in non-recurrent bladder cancer patients [28]. Accordingly, methylation profiling of urine sediments, to detect the top four frequently methylated genes, namely *SALL3*, *CFTR*, *ABCC6*, and *HPP1*, together can detect bladder cancer with 82.6% sensitivity and 100% specificity [29]. The *SALL3* gene might thus play a role in the tumorigenesis and could serve as an important biomarker for human cancers (Table 4).

In this study, we present a more comprehensive epigenetic analysis of chromosome 18q23 including its methylation patterns in HNSCC and normal mucosal tissues. Our study is the first to investigate the pattern of DNA methylation at chromosome 18q23, as well as its relationship with poor survival and early relapse. Regarding cancer risk, we also showed that there might also be a role for DNA methylation at this important cancer susceptibility locus.

## 4. Methods

### 4.1. Tumor Samples of Original Cohort

In total, 243 primary HNSCC samples were obtained from patients during surgery at the Department of Otolaryngology, Hamamatsu University School of Medicine. The samples were obtained soon after diagnosis and were thus from untreated tumors. Pertinent information including age, sex, smoking status, alcohol consumption, primary tumor site, tumor size, lymph node status, and clinical stage was obtained from the patients’ medical records. The male:female ratio in the patient cohort was 208:35. The mean age was 65.1 years (range, 32–92 years). Primary tumors were in the hypopharynx (*n* = 58), larynx (*n* = 47), oropharynx (*n* = 66), or oral cavity (*n* = 72). All patients provided written informed consent, and the study protocol was approved by the Institutional Review Board of the Hamamatsu University School of Medicine (date of board approval: 2 October 2015, ethic code: 25-149).

### 4.2. Target Gene Selection

Examination of the number and size of CpG islands and the density of CpG sites in upstream and downstream flanking sequences of the transcription start sites of ADNP homeobox 2 (*ADNP2*), myelin basic protein (*MBP*), small integral membrane protein 21 (*SMIM21*), and zinc finger protein 407 (*ZNF407*) was eliminated from the study targets since they did not contain CpG islands within the target regions. We therefore selected 12 genes located on 18q23 (*ATP9B*, *CTDP1*, *GALR1*, *HSBP1L1, KCNG2*, *NFATC1*, *PARD6G*, *PQLC1*, *RBFA*, *SALL3*, *TXNL4A* and *ZNF516*), which had CpG islands in their transcription start sites. A CpG island is defined as a DNA segment fulfilling the following three conditions: (i) length of the segment is at least 200 bp, (ii) G and C contents are ≥50%, and (iii) the observed CpG to expected CpG ratio is ≥0.6 [30].

### 4.3. Q-MSP Analysis

Extraction and bisulfite conversion of genomic DNA from 243 primary HNSCC and 36 noncancerous mucosal samples were performed using the MethylEasy Xceed Rapid DNA Bisulfite Modification Kit (TaKaRa, Tokyo, Japan), as per the manufacturer’s instructions [31,32]. The bisulfite-modified DNA was used as a template for fluorescence-based real-time PCR [2]. The methylation levels of the CpG islands in the promoters of the 12 genes located on 18q23 were determined via Q-MSP with the TaKaRa Thermal Cycler Dice Real Time System TP800 (TaKaRa). A list of the primer sequences for Q-MSP analysis is shown in Appendix A. Exon structures and CpG sites within the expanded views of the promoter region relative to the transcription start site are presented in Appendix A. A standard curve was constructed by plotting known concentrations of serially-diluted EpiScope Methylated HeLa gDNA (TaKaRa). NMVs were determined as follows: NMV = (target gene-S/target gene-FM)/(ACTB-S/ACTB-FM), where target gene-S and target gene-FM represent target gene methylation levels in the tumor sample and universal methylated DNA control, respectively, and ACTB-S and ACTB-FM represent *ACTB* (which encodes β-actin) methylation levels in the sample and control, respectively. For amplification reactions, 2 μL (0.01 μg/μL) of bisulfite-treated genomic DNA, 12.5 μL of SYBR^®^ Premix DimerEraser TM Perfect Real Time (TaKaRa), and 0.5 μL (10 μM) of each primer were added to a final volume of 25 μL. The PCR conditions were as follows: one denaturing cycle at 95 °C for 10 s, followed by 40 cycles of denaturing at 95 °C for 5 s and annealing/extension at 58 °C for 30 s (two-step reaction). Dissociation curves were carried out at the end of each PCR by following a 3-step procedure. Analysis was performed using the software (version 1.03A) for the Thermal Cycler Dice Real Time System TP800 (TaKaRa), according to the manufacturer’s directions [33,34].

### 4.4. Analysis of HPV Status

To assess HPV status, samples were also subjected to PCR using specific primers for HPV types 16, 18, 31, 33, 35, 52 and 58. The PCR HPV Typing Set (TaKaRa) method was performed according to the manufacturer’s protocol. The PCR products were separated using 9% polyacrylamide gel electrophoresis followed by ethidium bromide staining.

### 4.5. Collection of Publicly Available Data from TCGA

Aberrant DNA methylation data available in TCGA (January 2019) were collected via the MethHC database (http://methhc.mbc.nctu.edu.tw/php/index.php) using the Infinium HumanMethylation450 platform (Illumina, Inc., San Diego, CA, USA) and were expressed as β values. The β value is a number between 0 (not methylated) and 1 (completely methylated) that represents the ratio of methylated allele intensity to overall intensity [35]. A conservative cut-off comprising β-values > the median values of normal TCGA samples was used to call samples as methylated. Cutoff levels were determined to be 0.029 for *ATP9B*, 0.143 for *GALR1*, 0.083 for *HSBP1L1*, 0.873 for *KCNG2*, 0.044 for *NFATC1*, 0.067 for *PARD6G* and 0.099 for *SALL3*.

### 4.6. Literature Review

A search of the PubMed database using the following terms was performed to identify studies reporting genes in which the detection of methylation or expression was significantly associated with their use as a biomarker for prognosis and diagnosis: “target gene name” AND “cancer” AND “survival” OR “diagnosis” (Table 4).

### 4.7. Data Analysis and Statistics

The Q-MSP results and patient characteristics (age of onset, sex, smoking status, alcohol consumption, HPV status, tumor size, lymph node status, clinical stage and recurrence events) were compared using a Student’s *t*-test. Receiver-operator characteristic (ROC) curve analysis was conducted using the NMVs for 36 HNSCC and 36 adjacent normal mucosal samples with the Stata/SE 13.0 system (Stata Corporation, College Station, TX, USA). Prediction accuracy was assessed using the area under the ROC curve. Cut-off values showing the greatest accuracy were determined based on sensitivity/specificity, as indicated in Appendix A. The cut-off values were used to determine the methylation frequencies of the target genes. The overall methylation rates in the individual samples were determined by calculating the MI. The MI was defined as the ratio of the number of methylated genes to the number of tested genes in each sample [34,36].

DFS was measured from the date of the initial treatment to the date of diagnosis of first recurrence, either locoregional or systemic. Survival curves were plotted using the Kaplan–Meier method and significance was assessed by the log-rank test. The prognostic value of methylation status was assessed by performing multivariate Cox proportional hazards analysis adjusting for age (≥70 versus <70 years), HPV status, smoking status, alcohol intake, and tumor stage (I, II and III versus IV). A *p*-value less than 0.05 was considered statistically significant. All statistical analyses were performed using StatMate IV software (IV version, ATMS Co. Ltd., Tokyo, Japan).

## 5. Conclusions

The current study provides evidence that detecting aberrant *GALR1* and *SALL3* methylation can serve as a means to identify critical events in HNSCC progression. As such, the promoter methylation status of 18q23 genes might be an important marker to explain distinct tumor patterns and behaviors with HNSCC. Our findings support the use of methylation markers to select patients for adjuvant therapy after initial surgical treatment; however, our preliminary findings need to be validated in larger and more homogeneous HNSCC patient cohorts.

## Figures and Tables

**Figure 1 cancers-11-00401-f001:**
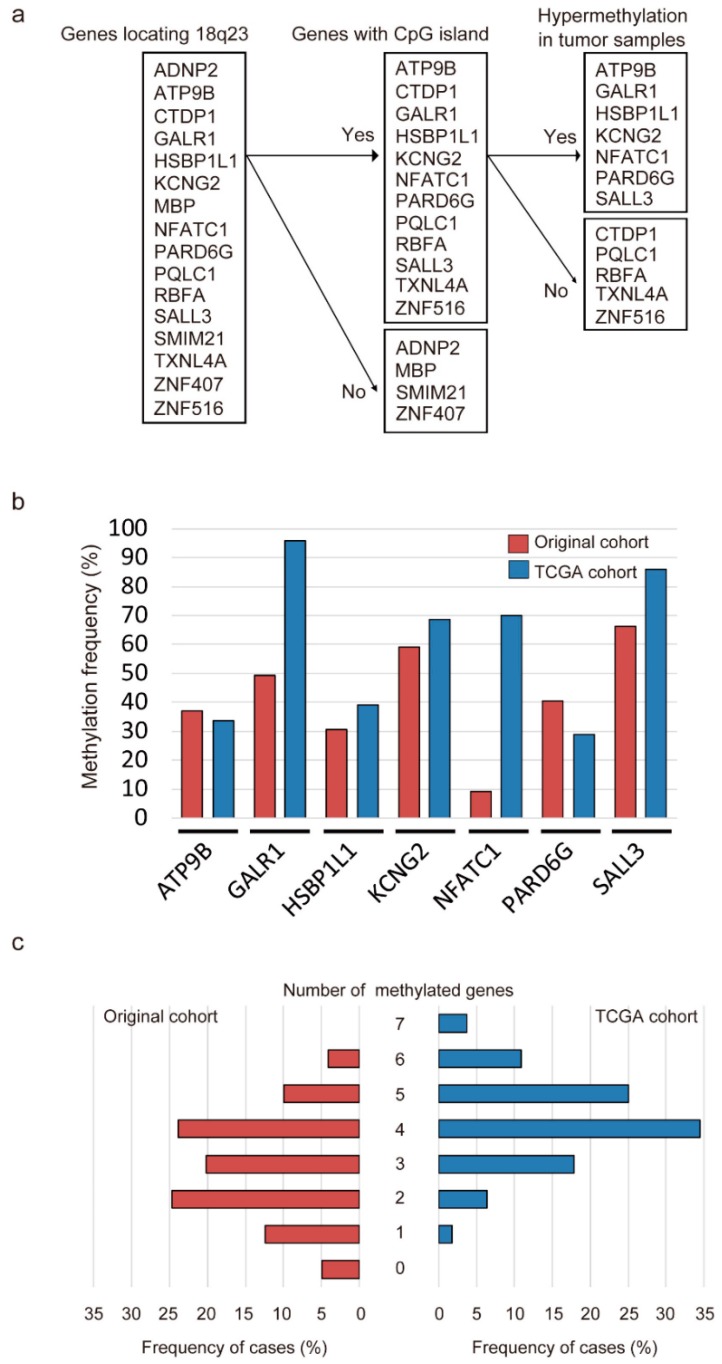
Methylation of genes located on 18q23 in head and neck squamous cell carcinoma (HNSCC) samples. (**a**) flow chart of the selection of candidate genes; (**b**) bar graph showing the methylation frequencies of the seven genes in the original cohort (red bars) and TCGA cohort (blue bars). (**c**) bar graph comparing the number of HNSCC cases to the number of methylated genes in the original cohort (red bars) and TCGA cohort (blue bars).

**Figure 2 cancers-11-00401-f002:**
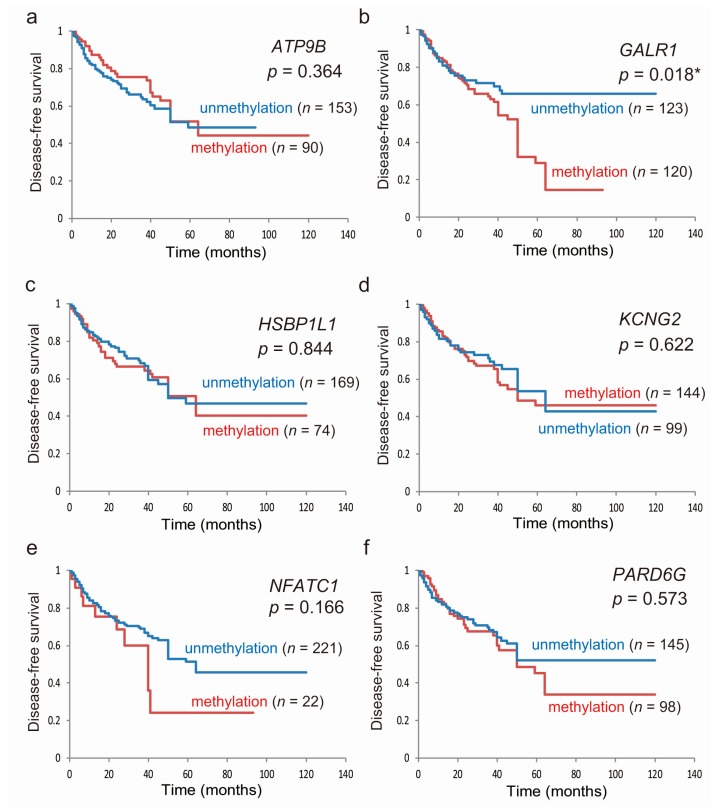
Kaplan–Meier survival curves for the 243 patients with HNSCC according to the methylation status of the seven target genes. Disease-free survival based on (**a**) ATP9B, (**b**) GALR1, (**c**) HSBP1L1, (**d**) KCNG2, (**e**) NFATC1, (**f**) PARD6G, and (**g**) SALL3 methylation; methylated (red lines) and unmethylated (blue lines) cases are shown; (**h**) joint analysis of GALR1 and SALL3 genes. blue line: patients with both unmethylated genes; green line: patients with either methylated gene; red line: patients with both methylated genes. A probability of <0.05 (* *p* < 0.05) was considered a statistically significant difference.

**Table 1 cancers-11-00401-t001:** Distribution of methylation status by selected epidemiologic and clinical characteristics in the original cohort.

Gene		ATP9B		GALR1		HSBP1L1		KCNG2		NFATC1		PARD6G		SALL3	
Characteristics	Methylation status	Yes	No	Yes	No	Yes	No	Yes	No	Yes	No	Yes	No	Yes	No
	Overall (%)	90 (37.0%)	153 (63.0%)	120 (49.4%)	123 (50.6%)	74 (30.5%)	169 (69.5%)	144 (59.3%)	99 (40.7%)	22 (9.1%)	221 (90.9%)	98 (40.3%)	145 (59.7%)	161 (66.3%)	82 (33.7%)
Age	<70	64	95	81	78	49	110	99	60	17	142	65	94	103	56
	>70	26	58	39	45	25	59	45	39	5	79	33	51	58	26
	*p* ^†^		0.165		0.59		0.658		1		0.25		0.891		0.569
Gender	Female	13	22	13	22	10	25	20	15	3	32	12	23	23	12
	Male	77	131	107	101	64	144	124	84	19	189	86	122	138	70
	*p* ^†^		1		0.144		0.846		1		1		0.463		1
Smoking status	Smoker	72	115	95	92	56	131	117	70	20	167	73	114	125	62
	Non-smoker	18	38	25	31	18	38	27	29	2	54	25	31	36	20
	*p* ^†^		0.433		0.449		1		0.064		0.118		1		1
Alcohol exposure	Drinker	68	116	94	90	54	130	112	72	19	165	76	108	126	58
	Non-drinker	22	37	26	33	20	39	32	27	3	56	22	37	35	24
	*p* ^†^		1		0.372		1		1		0.301		0.649		1
HPV status	Positive	17	29	28	18	9	37	28	18	5	41	23	23	34	12
	Negative	73	124	92	105	65	132	116	81	17	180	75	122	127	70
	*p* ^†^		1		0.102		0.078		0.869		1		1		0.299
Tumor size	T1–2	47	75	56	66	37	85	68	54	11	111	44	78	86	36
	T3–4	43	78	64	57	37	84	76	45	11	110	54	67	75	46
	*p* ^†^		0.691		0.306		1		0.3		1		0.192		0.176
Lympho-node status	N0	33	65	46	52	27	71	58	40	6	92	34	64	63	35
	N+	57	88	74	71	47	98	86	59	16	129	64	81	98	47
	*p* ^†^		0.417		0.601		0.478		1		0.255		0.146		1
Stage	I–III	37	67	47	57	31	73	61	43	5	99	37	67	67	37
	IV	53	86	73	66	43	96	83	56	17	122	61	78	94	45
	*p* ^†^		0.69		0.3		0.889		1		0.069		0.234		1
Recurrence events	Positive	31	54	52	33	32	53	50	35	10	75	35	50	64	21
	Negative	59	99	68	90	42	116	94	64	12	146	63	95	97	61
	*p* ^†^		1		0.007 *		1		1		1		1		0.033 *

^†^ Chi-squared test, * *p* < 0.05.

**Table 2 cancers-11-00401-t002:** Distribution of methylation status by selected epidemiologic and clinical characteristics in the TCGA cohort.

Gene		ATP9B		GALR1		HSBP1L1		KCNG2		NFATC1		PARD6G		SALL3	
Characteristics	Methylation status	Yes	No	Yes	No	Yes	No	Yes	No	Yes	No	Yes	No	Yes	No
	Overall(%)	174 (33.7%)	342 (66.3%)	495 (95.9%)	21 (4.1%))	201 (39.0%)	315 (61.0%)	354 (68.6%)	162 (31.4%)	361 (70.0%)	155 (30.0%)	149 (28.9%)	367 (71.1%)	443 (85.9%)	73 (14.1%)
Age	<70	142	258	383	17	154	246	269	131	286	114	114	286	337	63
	>70	31	84	111	4	47	68	84	31	74	41	34	81	105	10
	*p* ^†^		0.094		1		1		0.256		1		1		0.068
Gender	Female	48	93	138	3	74	67	86	55	92	49	46	95	125	16
	Male	126	249	357	18	127	248	268	107	269	106	103	272	318	57
	*p* ^†^		1		0.216		<0.001 *		1		1		1		0.321
Smoking status	Smoker	132	250	367	15	135	247	269	113	271	111	118	264	326	56
	Non-smoker	39	83	116	6	63	59	77	45	85	37	31	91	107	15
	*p* ^†^		0.661		1		0.002 *		1		1		0.257		0.554
Alcohol exposure	Drinker	107	237	329	15	140	204	233	111	238	106	106	238	295	49
	Non-drinker	61	100	155	6	55	106	114	47	113	48	41	120	138	23
	*p* ^†^		1		0.816		0.171		0.537		0.837		0.248		1
Tumor size	T1–2	58	150	200	8	80	128	143	65	142	66	68	140	181	27
	T3–4	116	192	295	13	121	187	211	97	219	89	81	227	262	46
	*p* ^†^		0.023 *		1		0.855		0.3		1		1		0.607
Lympho-node status	N0	67	155	218	4	95	127	145	77	150	72	66	156	195	27
	N+	106	180	269	17	101	185	203	83	207	79	83	203	240	46
	*p* ^†^		0.11		0.024 *		1		1		1		1		0.251
Stage	I–III	56	153	203	6	90	119	149	60	145	64	67	142	182	27
	IV	118	189	292	15	111	196	205	102	216	91	82	225	261	46
	*p* ^†^		0.006 *		0.364		1		0.289		1		1		0.524

^†^ Chi-squared test, * *p* < 0.05.

**Table 3 cancers-11-00401-t003:** Methylation status of individual genes and associations with disease-free survival using Cox proportional hazards model in 243 patients.

Gene	Methylation Status	Overall (%)	Recurrence Events		Adjusted RR (95% CI) ^†^
			Positive (*n* = 85)	Negative (*n* = 158)	
ATP9B	Yes	90 (37.0%)	31	59	
	No	153 (63.0%)	54	99	0.831 (0.531–1.301)
GALR1	Yes	120 (49.4%)	52	68	
	No	123 (50.6%)	33	90	1.600 (1.027–2.493) *
HSBP1L1	Yes	74 (30.5%)	32	42	
	No	169 (69.5%)	53	116	1.016 (0.644–1.603)
KCNG2	Yes	144 (59.3%)	50	94	
	No	99 (40.7%)	35	64	1.199 (0.761–1.890)
NFATC1	Yes	22 (9.1%)	10	12	
	No	221 (90.9%)	75	146	1.455 (0.731–2.897)
PARD6G	Yes	98 (40.3%)	35	63	
	No	145 (59.7%)	50	95	1.116 (0.719–1.731)
SALL3	Yes	161 (66.3%)	64	97	
	No	82 (33.7%)	21	61	1.911 (1.155–3.162) *
MI	5–6	34 (14.0%)	17	17	
	0–4	209 (86.0%)	68	141	1.609 (0.932–2.778)

^†^ Adjusted for age, gender, smoking status, alcohol exposure and stage. * *p* < 0.05; CI: confidence interval. RR: recurrence ratio.

**Table 4 cancers-11-00401-t004:** Published studies of 18q23 genes and biomarkers for patients.

Genes Studied	Study (Ref.)	Year	Country	Cases	Disease	Significant Association with Survival and Diagnosis
GALR1 hypermethylation	Kanazawa T et al. [19]	2018	Japan	34	salivary duct carcinoma	Worse survival (*p* = 0.026)
GALR1 hypermethylation	Doufekas K et al. [20]	2013	United Kingdom	64	endometrial cancer	Vaginal swabs for detection of endometrial cancer; a sensitivity of 92.7% and a specificity of 78.9%
NFATC1 hypomethylation	Wolf C et al. [21]	2018	Germany	130	Chronic lymphocytic leukemia	Disease progression stages (*p* < 0.05)
NFATC1 overexpression	Kawahara T et al. [22]	2017	Japan	99	urothelial carcinoma	Lower progression-free survival (*p* = 0.032)
NFATC1 overexpression	Li L et al. [23]	2016	China	93	ovarian cancer	Worse overall survival (*p* < 0.01)
SALL3 hypermethylation	Heijden AG et al. [28]	2018	Spain	458	bladder cancer	Higher recurrence rate (*p* < 0.001)
SALL3 hypermethylation	Wei X et al. [27]	2015	China	23	cervical cancer	HPV infection positive relationship (*p* = 0.010, *r* = 0.408)
SALL3 hypermethylation	Yu J et al. [29]	2007	China	132	bladder cancer	Urine sediments for detection of bladder cancer; a sensitivity of 58.3% and a specificity of 100%

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
