# Peer review of "Genes Located on 18q23 Are Epigenetic Markers and Have Prognostic Significance for Patients with Head and Neck Cancer"

_cancers, 2019, doi:10.3390/cancers11030401_

Round 1
Reviewer 1 Report
This is a well-constructed manuscript by the authors which shows the prognostic significance of 18q23 methylation in HNSCC. The authors did a detailed analysis of 243 different tumors using promoter methylation analysis and compared results to the TCGA data. Statistical methods are appropriate for the analysis and COX models show significance of the results. The potential tumor suppressive role of the different genes at this locus is of interest. My primary question is using the TCGA data set that was also analyzed, how does the expression of the different genes in this locus correlate with methylation status in the tumors?
Author Response
February 25th, 2019
Dr. Angeliki Magklara and Dr. Tugba Bagci-Onder
Special Issue Editor
Cancers
Dear Editor: Dr. Angeliki Magklara and Dr. Tugba Bagci-Onder
We appreciate the opportunity to revise our manuscript entitled “Genes located on 18q23 are epigenetic markers and have prognostic significance for patients with head and neck cancer.” (Manuscript ID, cancers-452927). We have included our point-by-point responses to the reviewers’ comments and questions below. All changes are highlighted in red in the revised manuscript. We hope that our manuscript is now acceptable for publication in the Cancers.
Sincerely,
Kiyoshi Misawa
Department of Otolaryngology/Head and Neck Surgery
Hamamatsu University School of Medicine
1-20-1 Handayama
Shizuoka 431-3192, Japan
Phone: 81-53-435-2252
Fax: 81-53-435-2253
E-mail: kiyoshim@hama-med.ac.jp
Dr. Angeliki Magklara and Dr. Tugba Bagci-Onder
Special Issue Editor
Cancers
Re: Manuscript ID; cancers-452927
Thank you for your letter dated February 21 that gave us the opportunity to revise our manuscript in response to the reviewers’ comments. We have provided point-by-point responses to the comments and questions from the reviewers below.
________________________________________
Reviewer 1
This is a well-constructed manuscript by the authors which shows the prognostic significance of 18q23 methylation in HNSCC. The authors did a detailed analysis of 243 different tumors using promoter methylation analysis and compared results to the TCGA data. Statistical methods are appropriate for the analysis and COX models show significance of the results. The potential tumor suppressive role of the different genes at this locus is of interest.
My primary question is using the TCGA data set that was also analyzed, how does the expression of the different genes in this locus correlate with methylation status in the tumors?
Our response: We appreciate the reviewer’s considered comments. We have analyzed the DNA methylation and mRNA levels in the TCGA cohort and have included the results (Figure S2) in the revised version of the manuscript.

Reviewer 2 Report
In this article, the authors perform a detailed analysis of the methylation status of the genes located at 18q23 on HNSCC using a retrospective cohort of their own patients and the data from the TCGA. From the 12 analyzed genes with CpG islands, 7 showed hypermethylation in HNSCC samples compared to normal tissue controls. These seven genes were also found to be hypermethylated in the TCGA cohort.
In addition, they found that GALR1 and SALL3 methylation were significantly associated with a higher frequency of tumor recurrence and lower disease-free survival. However, the analysis of the TCGA cohort did not confirm the prognostic significance of the methylation of these genes. In fact, although not significant, in the TCGA cohort, the cases with methylation of the genes at 18q23 showed better overall survival.
Based in their findings, in the conclusions they claim that “The current study provides evidence that the methylation status of SALL3 and GALR1 is an independent prognostic factor for DFS in patients with HNSCC”. But these results were only found in their cohort and not confirmed in the analysis of the TCGA data, and therefore lack the claimed evidence.
Specific comments:
- In the introduction, they stated that “to our knowledge, this study is the first to implicate 18q23 gene methylation in the genesis of HNSCC”. However, they simply describe the methylation status of the genes located at this locus in HNSCCs, and do not describe the mechanisms by which the methylation of these genes is involved in the genesis of these tumors.
- On results, point 2.4, they mentioned that “Based on log-rank tests, a trend in poorer overall survival for patients with the methylation phenotype, defined as ≥ 5 methylated genes, was observed (P = 0.056; Table S1)”. However the analysis indicated in the table and methods id for disease-free survival.
- In the discussion (line 163) they mention again that “To our knowledge, this study is the first to show that the epigenetic regulation of 18q23 genes can provide insights into the mechanisms of tumorigenesis and the risk of disease recurrence in HNSCC.” However, the work does not contain mechanistic studies by which epigenetic silencing of genes located in 18q23 may be implicated in the genesis of head and neck carcinomas. Nor is there any confirmation in the TCGA data that its methylation has prognostic value.
- The table 4 includes a selection of studies that show the implication of 18q23 genes in several malignancies and its potential use as biomarkers. But this could not be considered a systematic review, as they do not show the criteria used to select the studies to be included in the table (negative studies could be omitted). A systematic review must follow the PRISMA guidelines.
- In figure S3, there must be a confusion with the time (1000 to 7000 months of follow-up?)
In summary, the study is correct, but the conclusions overestimate the importance of the findings, especially the prognostic value, which could not be confirmed.
Author Response
February 25th, 2019
Dr. Angeliki Magklara and Dr. Tugba Bagci-Onder
Special Issue Editor
Cancers
Dear Editor: Dr. Angeliki Magklara and Dr. Tugba Bagci-Onder
We appreciate the opportunity to revise our manuscript entitled “Genes located on 18q23 are epigenetic markers and have prognostic significance for patients with head and neck cancer.” (Manuscript ID, cancers-452927). We have included our point-by-point responses to the reviewers’ comments and questions below. All changes are highlighted in red in the revised manuscript. We hope that our manuscript is now acceptable for publication in the Cancers.
Sincerely,
Kiyoshi Misawa
Department of Otolaryngology/Head and Neck Surgery
Hamamatsu University School of Medicine
1-20-1 Handayama
Shizuoka 431-3192, Japan
Phone: 81-53-435-2252
Fax: 81-53-435-2253
E-mail: kiyoshim@hama-med.ac.jp
Dr. Angeliki Magklara and Dr. Tugba Bagci-Onder
Special Issue Editor
Cancers
Re: Manuscript ID; cancers-452927
Thank you for your letter dated February 21 that gave us the opportunity to revise our manuscript in response to the reviewers’ comments. We have provided point-by-point responses to the comments and questions from the reviewers below.
________________________________________
Reviewer 2
In this article, the authors perform a detailed analysis of the methylation status of the genes located at 18q23 on HNSCC using a retrospective cohort of their own patients and the data from the TCGA. From the 12 analyzed genes with CpG islands, 7 showed hypermethylation in HNSCC samples compared to normal tissue controls. These seven genes were also found to be hypermethylated in the TCGA cohort.
In addition, they found that GALR1 and SALL3 methylation were significantly associated with a higher frequency of tumor recurrence and lower disease-free survival. However, the analysis of the TCGA cohort did not confirm the prognostic significance of the methylation of these genes. In fact, although not significant, in the TCGA cohort, the cases with methylation of the genes at 18q23 showed better overall survival.
Based in their findings, in the conclusions they claim that “The current study provides evidence that the methylation status of SALL3 and GALR1 is an independent prognostic factor for DFS in patients with HNSCC”. But these results were only found in their cohort and not confirmed in the analysis of the TCGA data, and therefore lack the claimed evidence.
Our Response: We thank the reviewer for the careful and positive review of our paper; we have responded to each point below and in the paper as indicated.
Specific comments:
- In the introduction, they stated that “to our knowledge, this study is the first to implicate 18q23 gene methylation in the genesis of HNSCC”. However, they simply describe the methylation status of the genes located at this locus in HNSCCs, and do not describe the mechanisms by which the methylation of these genes is involved in the genesis of these tumors.
Our response: Thank you for your comments. On page 2, line 70, we have revised the sentence as follows: to our knowledge, this study is the first to implicate 18q23 gene methylation in the analysis of HNSCC.
- On results, point 2.4, they mentioned that “Based on log-rank tests, a trend in poorer overall survival for patients with the methylation phenotype, defined as ≥ 5 methylated genes, was observed (P = 0.056; Table S1)”. However, the analysis indicated in the table and methods id for disease-free survival.
Our response: Thank you for your careful review. On page 7, line 135, we have revised the sentence as follows: Based on log-rank tests, a trend in poorer DFS for patients with the methylation phenotype, defined as ≥ 5 methylated genes, was observed (P = 0.056; Table S1).
- In the discussion (line 163) they mention again that “To our knowledge, this study is the first to show that the epigenetic regulation of 18q23 genes can provide insights into the mechanisms of tumorigenesis and the risk of disease recurrence in HNSCC.” However, the work does not contain mechanistic studies by which epigenetic silencing of genes located in 18q23 may be implicated in the genesis of head and neck carcinomas. Nor is there any confirmation in the TCGA data that its methylation has prognostic value.
Our response: Thank you for your comments. On page 9, line 166, we have revised the sentence as follows: To our knowledge, this study is the first to show that the epigenetic regulation of 18q23 genes can provide insights into the aggressive tumor behavior and the risk of disease recurrence in HNSCC.
- The table 4 includes a selection of studies that show the implication of 18q23 genes in several malignancies and its potential use as biomarkers. But this could not be considered a systematic review, as they do not show the criteria used to select the studies to be included in the table (negative studies could be omitted). A systematic review must follow the PRISMA guidelines.
Our response: I agree with your comment. In the revised methods, we have changed “Systematic literature review” to “Literature review”.
- In figure S3, there must be a confusion with the time (1000 to 7000 months of follow-up?)
Our response: Thank you for your careful review. In Figure S3, we have changed the months of follow-up to the days of follow-up.
In summary, the study is correct, but the conclusions overestimate the importance of the findings, especially the prognostic value, which could not be confirmed.
Our response: Thank you for this suggestion. On page 13, line 303, we have revised the sentence as follows: As such, the promoter methylation status of 18q23 genes might be an important marker to explain distinct tumor patterns and behaviors with HNSCC. Our findings support the use of methylation markers in patient selection for adjuvant therapy after initial surgical treatment; however, our preliminary findings need to be validated in larger and more homogeneous HNSCC patient cohorts.

Reviewer 3 Report
The manuscript by Misawa et al. presents the results of identification of two genes those methylation status might be important for the tumorigenesis and the risk of disease recurrence in HNSCC. And although the work is interesting to some extent, it does show features of preliminary study.
Major concern:
The main problem of the work is limiting the experiments to actually one method (Q-MSP), which despite the fact that it can work well in some area, has also its disadvantages and limitations. Thus, the authors do not even try to validate the real values of their results by applying a different methodology.
Minor concerns:
1. The abbreviations should be explained when they appear for the first time (e.g. Line 76).
2. For better presentation authors should place a schematic draw of each gene, mark the methylation islands, their size, and distance in relation to ATG initiation codon. It is not clear how they identified the TSSs?
3. Authors should explain what methodology they used for the identification of CpG island and write how they define CpG island.
4. Section 2.2. Lines 92-97. There is huge differences in methylation status of GALR1 and NFATC1 genes between two cohorts (49.4% vs. 95.9% and 9.1% vs. 70% respectively). Authors should discuss it and explain these differences.
5. Discussion section. In general discussion is poorly written and contains many statements semantically empty, for example, the authors claim Lines 163-164: “To our knowledge, this study is the first to show that the epigenetic regulation of 18q23 genes can provide insights into the mechanisms of tumorigenesis and the risk of disease recurrence in HNSCC.” Please explain how these results provide any mechanistical insights into tumorigenesis of HNSCC?
6. Authors do not discuss the statistical significant results for the HSBP1L1 and ATP9B (Table 2). It is not clear why?
7. Methods section. Q-MSP is poorly described and makes it impossible to repeat the analysis. Authors should in detail describe the methodology and provide how DNA was processed, how many ng of DNA was analysed per sample, what were the conditions of the PCR etc.
Author Response
February 25th, 2019
Dr. Angeliki Magklara and Dr. Tugba Bagci-Onder
Special Issue Editor
Cancers
Dear Editor: Dr. Angeliki Magklara and Dr. Tugba Bagci-Onder
We appreciate the opportunity to revise our manuscript entitled “Genes located on 18q23 are epigenetic markers and have prognostic significance for patients with head and neck cancer.” (Manuscript ID, cancers-452927). We have included our point-by-point responses to the reviewers’ comments and questions below. All changes are highlighted in red in the revised manuscript. We hope that our manuscript is now acceptable for publication in the Cancers.
Sincerely,
Kiyoshi Misawa
Department of Otolaryngology/Head and Neck Surgery
Hamamatsu University School of Medicine
1-20-1 Handayama
Shizuoka 431-3192, Japan
Phone: 81-53-435-2252
Fax: 81-53-435-2253
E-mail: kiyoshim@hama-med.ac.jp
Dr. Angeliki Magklara and Dr. Tugba Bagci-Onder
Special Issue Editor
Cancers
Re: Manuscript ID; cancers-452927
Thank you for your letter dated February 21 that gave us the opportunity to revise our manuscript in response to the reviewers’ comments. We have provided point-by-point responses to the comments and questions from the reviewers below.
________________________________________
Reviewer 3
The manuscript by Misawa et al. presents the results of identification of two genes those methylation statuses might be important for the tumorigenesis and the risk of disease recurrence in HNSCC. And although the work is interesting to some extent, it does show features of preliminary study.
Major concern:
The main problem of the work is limiting the experiments to one method (Q-MSP), which despite the fact that it can work well in some area, has also its disadvantages and limitations. Thus, the authors do not even try to validate the real values of their results by applying a different methodology.
Our response: We thank the reviewer for the positive review of our paper. We fully agree with this comment. We have analyzed the DNA methylation and mRNA levels in the TCGA cohort and have included the results (Figure S2) in the revised version of the manuscript.
Minor concerns:
1. The abbreviations should be explained when they appear for the first time (e.g. Line 76).
Our response: Thank you for your careful comments. On page 2, line 76, the abbreviations errors have been corrected; e.g. Normalized methylation values (NMVs).
2. For better presentation authors should place a schematic draw of each gene, mark the methylation islands, their size, and distance in relation to ATG initiation codon. It is not clear how they identified the TSSs?
Our response: Thank you for your comments. We have added a schematic representation of CpG sites within expanded views of the promoter region relative to the TSS. We have added this information as Figure S5.
3. Authors should explain what methodology they used for the identification of CpG island and write how they define CpG island.
Our response: The reviewer makes a good point. On page 11, line 240, we have added the following information; CpG island is defined as a DNA segment fulfilling the following three conditions: (i) length of segment is at least 200 bp, (ii) G and C contents are ≥ 50%, and (iii) observed CpG to expected CpG ratio (o/e) is ≥ 0.6. (Gardiner-Garden M, Frommer M. CpG islands in vertebrate genomes. J. Mol. Biol. 1987;196(2):261.)
4. Section 2.2. Lines 92-97. There is huge differences in methylation status of GALR1 and NFATC1 genes between two cohorts (49.4% vs. 95.9% and 9.1% vs. 70% respectively). Authors should discuss it and explain these differences.
Our response: The reviewer makes a good point. GALR1 and NFATC1 genes have huge differences between normalized methylation values (original cohort) and β-values (TCGA cohort). However, other five genes (ATP9B, HSBP1L1, KCHG2, PARD6G and SALL3) are tiny differences between two values. We conclude that these differences are statistical strategy between Q-MSP method and Infinium HumanMethylation450 platform method.
5. Discussion section. In general discussion is poorly written and contains many statements semantically empty, for example, the authors claim Lines 163-164: “To our knowledge, this study is the first to show that the epigenetic regulation of 18q23 genes can provide insights into the mechanisms of tumorigenesis and the risk of disease recurrence in HNSCC.” Please explain how these results provide any mechanistical insights into tumorigenesis of HNSCC?
Our response: Thank you for your comments. On page 9, line 166, we have revised the sentence as follows: To our knowledge, this study is the first to show that the epigenetic regulation of 18q23 genes can provide insights into the aggressive tumor behavior and the risk of disease recurrence in HNSCC.
6. Authors do not discuss the statistical significant results for the HSBP1L1 and ATP9B (Table 2). It is not clear why?
Our response: The reviewer makes a good point. In the TCGA cohort, methylation of the ATP9B promoter was higher in patients with advance stage, and methylation of the HSBP1L1 promoter was higher in patients with female and non-smoker. It is very interesting results, but we don’t have enough information and background materials about ATP9B and HSBP1L1
7. Methods section. Q-MSP is poorly described and makes it impossible to repeat the analysis. Authors should in detail describe the methodology and provide how DNA was processed, how many ng of DNA was analysed per sample, what were the conditions of the PCR etc.
Our response: On page 12, line 257, we have added the following information; For amplification reactions, 2 μL (0.01 μg/μL) of bisulfite treatment of genomic DNA, 12.5 μL of SYBR® Premix DimerEraser TM Perfect Real Time (TaKaRa), and 0.5 μL (10 μM) of each primer were added to a final volume of 25 μL. The PCR conditions were as follows: one denaturing cycle at 95 °C for 10 s, followed by 40 cycles of denaturing at 95 °C for 5 s, and annealing/extension at 58 °C for 30 s (two-step reaction). Dissociation curves are carried out at the end of a PCR experiment by following a 3-step procedure.

Reviewer 4 Report
Authors set out to assess the hypermethylation of genes located in 18q23 regions. The genes GALR1 and SALL3 promoters show highest methylation frequencies among 7 genes in both original and TCGA cohorts. The authors found that aberrant GALR1 and SALL3 promoter methylaiton correlates with disease recurrence in-patient with HNSCC. Authors used a cohort of 243 patients and TCGA cases for these studies. Epigenetic repression of SALL3 is shown previously as major predictor of DFS. Additionally, GALR1 has been shown to be tumor suppressor and this phenotype in part is mediated by promoter methylation. The meta analyses confirm their previous findings. The works seems combine both studies using metadata.
Minor: 1. Line 61 need reference: https://clinicalepigeneticsjournal.biomedcentral.com/articles/10.1186/s13148-017-0363-1
Line 70??
Author Response
February 25th, 2019
Dr. Angeliki Magklara and Dr. Tugba Bagci-Onder
Special Issue Editor
Cancers
Dear Editor: Dr. Angeliki Magklara and Dr. Tugba Bagci-Onder
We appreciate the opportunity to revise our manuscript entitled “Genes located on 18q23 are epigenetic markers and have prognostic significance for patients with head and neck cancer.” (Manuscript ID, cancers-452927). We have included our point-by-point responses to the reviewers’ comments and questions below. All changes are highlighted in red in the revised manuscript. We hope that our manuscript is now acceptable for publication in the Cancers.
Sincerely,
Kiyoshi Misawa
Department of Otolaryngology/Head and Neck Surgery
Hamamatsu University School of Medicine
1-20-1 Handayama
Shizuoka 431-3192, Japan
Phone: 81-53-435-2252
Fax: 81-53-435-2253
E-mail: kiyoshim@hama-med.ac.jp
Dr. Angeliki Magklara and Dr. Tugba Bagci-Onder
Special Issue Editor
Cancers
Re: Manuscript ID; cancers-452927
Thank you for your letter dated February 21 that gave us the opportunity to revise our manuscript in response to the reviewers’ comments. We have provided point-by-point responses to the comments and questions from the reviewers below.
______________________________________
Reviewer 4
Authors set out to assess the hypermethylation of genes located in 18q23 regions. The genes GALR1 and SALL3 promoters show highest methylation frequencies among 7 genes in both original and TCGA cohorts. The authors found that aberrant GALR1 and SALL3 promoter methylation correlates with disease recurrence in-patient with HNSCC. Authors used a cohort of 243 patients and TCGA cases for these studies. Epigenetic repression of SALL3 is shown previously as major predictor of DFS. Additionally, GALR1 has been shown to be tumor suppressor and this phenotype in part is mediated by promoter methylation. The meta analyses confirm their previous findings. The works seems combine both studies using metadata.
Minor: 1. Line 61 need reference: https://clinicalepigeneticsjournal.biomedcentral.com/articles/10.1186/s13148-017-0363-1
Our response: Thank you for your comments. On Page 2: line 61, I added the reference: Misawa, K.; Mochizuki, D.; Imai, A.; Misawa, Y.; Endo, S.; Mima, M.; Kawasaki, H.; Carey, T.E.; Kanazawa, T. Epigenetic silencing of sall3 is an independent predictor of poor survival in head and neck cancer. Clinical epigenetics 2017, 9, 64.
Line 70??
Our response: Thank you for your comments. Other reviewers gave the same instructions. On page 2, line 70, we have revised the sentence as follows: to our knowledge, this study is the first to implicate 18q23 gene methylation in the analysis of HNSCC.

Round 2
Reviewer 3 Report
In general, authors had addressed majority of my concerns. The most important points
were dismissed in an unsatisfactory fashion
Major concern:
The main problem of the work is limiting the experiments to one method (Q-MSP), which despite the fact that it can work well in some area, has also its disadvantages and limitations. Thus, the authors do not even try to validate the real values of their results by applying a different methodology.
Our response: We thank the reviewer for the positive review of our paper. We fully agree with this comment. We have analyzed the DNA methylation and mRNA levels in the TCGA cohort and have included the results (Figure S2) in the revised version of the manuscript.
I could accept the effort of the authors if they analyzed the expression of selected transcripts in their own cohort, but they did extract data from the existing database. It is hard to call it experimental validation of Q-MSP.
4. Section 2.2. Lines 92-97. There is huge differences in methylation status of GALR1 and NFATC1 genes between two cohorts (49.4% vs. 95.9% and 9.1% vs. 70% respectively). Authors should discuss it and explain these differences.
Our response: The reviewer makes a good point. GALR1 and NFATC1 genes have huge differences between normalized methylation values (original cohort) and β-values (TCGA cohort). However, other five genes (ATP9B, HSBP1L1, KCHG2, PARD6G and SALL3) are tiny differences between two values. We conclude that these differences are statistical strategy between Q-MSP method and Infinium HumanMethylation450 platform method.
It is difficult to accept this explanation. I do not understand how the statistical strategy may influence the experimental values of methylation levels? Especially in the context of other genes whose methylation levels in both cohorts are similar. Or maybe using another experimental method in authors' cohort (e.g. HRM) would help? The authors must explain this in a convincing way and discuss it in the article's text.
Please describe the conditions for a 3-step procedure for dissociation curves.
Author Response
February 28th, 2019
Dr. Angeliki Magklara and Dr. Tugba Bagci-Onder
Special Issue Editor
Cancers
Dear Editor: Dr. Angeliki Magklara and Dr. Tugba Bagci-Onder
We appreciate the opportunity to revise our manuscript entitled “Genes located on 18q23 are epigenetic markers and have prognostic significance for patients with head and neck cancer.” (Manuscript ID, cancers-452927). We have included our point-by-point responses to the reviewers’ comments and questions below. All changes are highlighted in red in the revised manuscript. We hope that our manuscript is now acceptable for publication in the Cancers.
Sincerely,
Kiyoshi Misawa
Department of Otolaryngology/Head and Neck Surgery
Hamamatsu University School of Medicine
1-20-1 Handayama
Shizuoka 431-3192, Japan
Phone: 81-53-435-2252
Fax: 81-53-435-2253
E-mail: kiyoshim@hama-med.ac.jp
In general, authors had addressed majority of my concerns. The most important points
were dismissed in an unsatisfactory fashion:
Major concern:
The main problem of the work is limiting the experiments to one method (Q-MSP), which despite the fact that it can work well in some area, has also its disadvantages and limitations. Thus, the authors do not even try to validate the real values of their results by applying a different methodology.
Our response: We thank the reviewer for the positive review of our paper. We fully agree with this comment. We have analyzed the DNA methylation and mRNA levels in the TCGA cohort and have included the results (Figure S2) in the revised version of the manuscript.
I could accept the effort of the authors if they analyzed the expression of selected transcripts in their own cohort, but they did extract data from the existing database. It is hard to call it experimental validation of Q-MSP.
4. Section 2.2. Lines 92-97. There is huge differences in methylation status of GALR1 and NFATC1 genes between two cohorts (49.4% vs. 95.9% and 9.1% vs. 70% respectively). Authors should discuss it and explain these differences.
Our response: The reviewer makes a good point. GALR1 and NFATC1 genes have huge differences between normalized methylation values (original cohort) and β-values (TCGA cohort). However, other five genes (ATP9B, HSBP1L1, KCHG2, PARD6G and SALL3) are tiny differences between two values. We conclude that these differences are statistical strategy between Q-MSP method and Infinium HumanMethylation450 platform method.
It is difficult to accept this explanation. I do not understand how the statistical strategy may influence the experimental values of methylation levels? Especially in the context of other genes whose methylation levels in both cohorts are similar. Or maybe using another experimental method in authors' cohort (e.g. HRM) would help? The authors must explain this in a convincing way and discuss it in the article's text.
Please describe the conditions for a 3-step procedure for dissociation curves.
Our response: Thank you for your comments. Your comments are very hard to quickly correspondence. We have a proposition for you. This article has two cohorts, our original cohort and TCGA cohort. If we present other original third cohort data of our laboratory, could we get your understanding?